# Single postoperative infusion of zoledronic acid to improve patient-reported outcome after hip or knee replacement: study protocol for a randomised, controlled, double-blinded clinical trial

Jonathan Brandt [ID],[1] Håkan Ledin,[1] Jonas Ranstam,[2] Ewa Roos [ID] ,[3] Per Aspenberg,[4,5] Jörg Schilcher[4,5,6]

Per Aspenberg died.

For numbered affiliations see end of article.

**Correspondence to**
Dr Jörg Schilcher;
jorg.schilcher@liu.se

## ABSTRACT

**Introduction** In Sweden, roughly 3000 patients are reoperated each year due to pain and loss of function related to a loosened hip or knee prosthesis. These reoperations are strenuous for the patient, technically demanding and costly for the healthcare system. Any such reoperation that can be prevented would be of great benefit. Bisphosphonates are drugs that inhibit osteoclast function. Several clinical trials suggest that bisphosphonates lead to improved implant fixation and one small study even indicates better functional outcome. Furthermore, in epidemiological studies, bisphosphonates have been shown to decrease the rate of revision for aseptic loosening by half. Thus, there are several indirect indications that bisphosphonates could improve patient-reported outcome, but no firm evidence.

**Methods and analysis** This is a pragmatic randomised, placebo-controlled, double-blinded, academic clinical trial of a single postoperative dose of zoledronic acid, in patients younger than 80 years undergoing primary total hip or knee replacement for osteoarthritis. Participants will be recruited from two orthopaedic departments. All surgeries will be performed, and study drugs given at Motala Hospital, Sweden. The primary endpoint is to investigate between-group differences in the Hip dysfunction and Osteoarthritis Outcome Score and the Knee injury and Osteoarthritis Outcome Score at 3-year follow-up. Secondary outcomes will be investigated at 1 year, 3 years and 6 years, and stratified for hip and knee implants. These secondary endpoints are supportive, exploratory or explanatory. A total of 1000 patients will be included in the study.

**Ethics and dissemination** The study has been approved by the Regional Ethical Review Board in Linköping (DNR 2015/286-31). The study will be reported in accordance with the Consolidated Standards of Reporting Trials statement for pharmacological trials. The results will be submitted for publication in peer-reviewed academic journals and disseminated to patient organisations and the media.

### Strengths and limitations of this study

► This is the first study to examine if a single intravenous dose zoledronic acid can improve patient-reported outcome after primary total hip or knee replacement.

► With 1000 patients included, this is the largest drug trial ever performed to test the effect of bisphosphonate treatment on the outcome after total joint replacement.

► The primary outcome variables, Hip dysfunction and Osteoarthritis Outcome Score and Knee injury and Osteoarthritis Outcome Score, are well validated and aim to directly evaluate patient-reported outcomes without the use of surrogate variables.

► Plain radiographs allow indirect assessment of treatment efficacy through radiographic evaluation of implant fixation as a secondary outcome.

► All patients are recruited from only two centres and operated at one single hospital, which might limit generalisability of the results.

**Trial registration number** EudraCT: No 2015-001200-55; Pre-results.

## INTRODUCTION

Total hip arthroplasty (THA) is one of the most successful operations ever invented and has been called 'The operation of the century'.[1] Total knee arthroplasty (TKA) has similar success rates.[2 3] When performed in elderly patients one can expect a less than 10% chance of ever needing secondary surgery.[4 5] However, in Sweden, roughly 3000 patients are operated on annually mainly because of pain and loss of function related to a loosened hip or knee prosthesis.[4] These reoperations are often difficult for the

surgeon and the patient, and the economic cost is several folds higher than for primary operation.[6] Also, results are less beneficial[7] and the complication rate is higher.[8] Any such reoperation that can be prevented would be of great benefit.

Implants loosen due to resorption of their bone bed by osteoclasts. When an implant is inserted into bone, a fracture healing response is activated.[9] This includes an increase in local bone formation and resorption, which are not necessarily coupled. If resorption outweighs bone formation, the initial fixation of the implant might be impaired. This excessive motion between the implant and its surrounding bone bed (implant migration) might allow pressurised fluid flows and invasion of wear debris particles leading to further bone resorption.[10] When direct bone contact does not occur in the early postoperative period, a fibrous tissue membrane will be formed leading to early subclinical loosening.[11 12] The primary postoperative result, that is, the fixation, can be estimated by specific radiographic methods (radiostereometry) to measure implant migration. There is a strong correlation between postoperative migration measured with radiostereometry and late loosening, showing that the early fixation is important for the late results. For acetabular cups, the area under the receiver operating characteristic curve for increased migration 2 years postoperatively to predict loosening after 10 years is 0.88 (95% CI 0.74 to 1.00).[13] Similar associations can be found for tibial components in total knee replacement.[14] This suggests that late loosening is the final result of a continuous process that starts immediately after the operation. Radiostereometry is partly invasive and very costly and can only be used in small series of patients.

Bisphosphonates specifically inhibit osteoclast activity, while in the fracture healing context bone formation remains increased. Therefore, bisphosphonate treatment at the time of implant insertion would possibly create a positive balance between bone formation and resorption leading to a net anabolic effect in the bone surrounding the implant thus leading to a more stable primary fixation.[15] Several randomised trials have shown that bisphosphonate treatment at the time of surgery reduces implant migration in TKA,[16] THA[17 18] and dental implants.[19] However, the effect of zoledronic acid on uncemented femoral stems remains unclear.[20] One clinical trial comprising a small sample of younger patients (n=50) also reported an improved functional outcome on the Harris Hip Score.[17] All other randomized controlled trials (RCT) showed no effect of bisphosphonate treatment on patient-reported outcome, but none of these trials were powered to detect such a difference. A recent meta-analysis of four epidemiological studies using hip and knee arthroplasty registries[21–24] has shown that bisphosphonate use is associated with a 50% decrease in the need for revision surgery.[25] Despite these findings, bisphosphonate treatment is not established in routine postoperative care to improve outcome after total joint replacement (TJR).

We here describe the study protocol for a pivotal trial designed to provide final evidence for the use of intravenous bisphosphonate to improve patient-reported outcome after primary THA and TKA.

## METHOD AND ANALYSIS
### Study design
This is a single centre, pragmatic, randomised, placebo-controlled, double-blinded, academic clinical trial. Participants will be recruited from two orthopaedic departments in Region Östergötland, Sweden. The main centre for recruitment is Motala Hospital (Capio Specialistvård Motala from 1 April 2019, and previously Aleris Specialistvård Motala), where roughly 85% of all patients will be recruited. The remaining 15% will be recruited from the Department of Orthopaedic Surgery at Linköping University Hospital. All patients will be referred to both orthopaedic departments based on standard healthcare routines in Region Östergötland. All surgeries will be performed, and study drugs given at Motala Hospital, Sweden. Patients scheduled for primary hip or knee arthroplasty, with respect given to inclusion and exclusion criteria, will be asked to participate both at the primary outpatient visit and after phone contact with a study nurse some weeks before the scheduled surgery. The final written consent will be given on the day of surgery (online supplemental file 1). All other treatment outside the study protocol described here will be according to the clinical routines of the hospital. Inclusion of patients was started on 4 January 2016. Data collection for the primary outcome will continue until patients have been followed for 3 years, roughly until 2024.

### Patients
One-thousand patients, 500 in each group, fulfilling the eligibility criteria box 1 will be included.

### Randomisation procedure and blinding
When found eligible, patients will be randomised to either zoledronic acid or placebo through block randomisation by the study nurse on the day of surgery. Block randomisation will be used to label infusion bags for drug delivery. The type of implant (hip or knee, cemented or not) will be a stratification factor in the randomisation to ensure balance among these factors.

All staff involved in patient care are blinded to treatment. The nurse in the postoperative care unit who is responsible for the preparation of the study drug according to the randomisation list will not be blinded. However, because this person is not otherwise involved in the study, concealment of treatment allocation is not jeopardised. The content of the infusion bag will be administered to the patient on the day after surgery by a blinded nurse in the surgical ward. The randomisation list will be available for unblinding in emergency situations 24-hours a day at Apoteksbolaget AB at Linköping University Hospital.

## Box 1  Overview of inclusion and exclusion criteria

**Inclusion criteria**

All patients eligible for primary hip or knee prosthesis for any form of osteoarthritis, between 18 years and 80 years of age

**Exclusion criteria**

Previous or present use of bisphosphonates or other antiresorptives

Present use of other drugs which influence bone, eg, anti-osteoporotic agents, glucocorticoids, anti-epileptics or use less than a year before randomisation

Present use of nephrotoxic medication

Active malignant disease

Pregnancy and breast feeding

Metabolic disease (other than osteoporosis) affecting the skeleton

Rheumatic disease

Hypocalcaemia as defined by local laboratory's criteria

Simultaneous bilateral surgery

Communication problems (drug abuse, language or behaviour problems)

Creatinine clearance, Glomerular filtration rate (GFR) <35 mL/min

Regular use of corticosteroids more than 5 mg prednisolone per day

Atypical fracture or osteonecrosis of the jaw

Expected follow-up period less than 3 years (eg, due to uncontrolled malignancy)

Expected to require special postoperative surveillance due to increased surgical risk (eg, for cardiac and psychiatric conditions)

### Intervention

Patients will be randomised to receive a single postoperative infusion of zoledronic acid 4 mg/5 mL[17] or placebo (5 mL saline) on the day after surgery.

### Study outcomes

#### Rationale for the outcome measures

In previous epidemiological studies of prosthetic loosening the endpoint has been revision surgery. We will report this parameter continuously and we will use the Swedish hip and knee arthroplasty registries to capture reoperations performed outside our uptake area. Since the overall revision rate for aseptic loosening in Sweden is around 2%–3% during a 10-year period, this endpoint would demand not only a very large study sample but also a long-term follow-up to get sufficient power. Also, some patients with loosening do not undergo revision surgery. They might be too old or fragile for these demanding operations. Other patients only have modest symptoms and might refrain from a demanding reoperation. Therefore, another primary outcome must be considered for reasons of feasibility. A previous study using the same treatment protocol of zoledronic acid as ours reported not only less implant migration in the zoledronic acid group but also a statistically significant improvement on the Harris Hip Score[17] two and 3 years postoperatively, despite small numbers (n=50). This study comprised only uncemented prostheses in younger patients with osteonecrosis of the femoral head. Based on these findings, because of the clinical importance and the predictive value on future revision surgery,[26–29] we have chosen to use patient-reported scores as our primary outcome: Hip dysfunction and Osteoarthritis Outcome Score

(HOOS), Swedish version LK 2.0[30] and the Knee injury and Osteoarthritis Outcome Score (KOOS), Swedish version LK 1.0.[31] Both instruments were meticulously designed with items generated in an iterative process, including input from stakeholder groups comprised of patients, orthopaedists and physical therapists. Both instruments have undergone extensive psychometric testing[32] and are recommended for evaluation of TJR by the International Consortium for Health Outcomes Measurement. These measures are free to use and have previously been shown to be highly reliable, with excellent internal consistency (Cronbach's alpha coefficient of 0.82–0.98) in samples of people undergoing THR and TKR.[33 34]

Our primary outcome measure will be between-group differences in KOOS/HOOS from baseline until the 3-year follow-up. Based on our literature review at the time of study design and confirmed later by the Outcome Measures in Rheumatology, workgroup TJR,[35] the subscale pain in HOOS/KOOS will be analysed as the primary endpoint in the confirmative analysis.

Secondary endpoints are included for supportive evidence. As secondary endpoints, we will analyse between-group differences in the remaining subscales of the KOOS/HOOS from baseline to 3 years, all subscales of the KOOS/HOOS and the RAND 36-Item Short Form Health Survey (RAND/SF-36) (Swedish version from 21 May 2013, using the 4-week recall period)[36 37] at 1 year, 3 years and 6 years and signs of radiographic loosening at 3 years and 6 years (table 1). The RAND/SF-36 will be analysed using physical and mental component scores.

### Statistical analysis

#### Power

Both HOOS and KOOS ranges from 0 to 100. The minimal important change is often reported to be 8–10 points, but this estimate is dependent on contextual factors such as patient's age, intervention and time to follow-up, and according to the developers of KOOS (www.koos.nu), no generic value of the minimal important change is available for KOOS, or any other patient-reported outcome measure. In a large sample, a small statistically significant difference is still indicative of an important treatment effect, even if the difference is smaller than the minimal clinically important difference. At the time of the study design, average reported KOOS values for 3 years after TKR were not available, and sample size calculations, therefore, were based on average values 2 years after TKR: 84 (SD 14). Lacking official consensus on a recommended clinically relevant difference, the research team decided on a 3-point difference on the HOOS/KOOS scale after 3 years. These values would with Student's t-test and a two-sided significance level of 5% yield 90% power when 450 patients are included in each arm. To compensate for a 10% withdrawal, a further 50 patients would be needed, leading to a total of 1000 patients to include in a superiority trial.

**Table 1** Schedule of assessments and events

| Visit | 1 | 2 | 3 | 4 | 5 | 6 | 7 | 8 |
|---|---|---|---|---|---|---|---|---|
| | Screening | Surgery | Treatment | Discharge | Follow-up | Follow-up | Follow-up | Follow-up |
| Time | Day −28 to day −2 | Day 1 | Day 2 | Day 3–5 | 6 weeks | 1 year ±1 month | 3 year ±1 month | 6 year ±1 month |
| **Assessment /event** | | | | | | | | |
| Informed consent* | X | | | | | | | |
| Demography medical history | X | | | | | | | |
| Physical examination | X | | | | | | | |
| Height and weight | X | | | | | | | |
| Assessment of inclusion and exclusion criteria | X | X | | | | | | |
| Blood sampling | X | | X | | | | | |
| Start of continuous daily calcium† | | | X | | | | | |
| Surgery | | X | | | | | | |
| Randomisation‡ | | X | X | | | | | |
| Administration of IMP/placebo§ | | X | X | | | | | |
| HOOS/KOOS RAND/SF-36 questionnaire | X | | | | | X | X | X |
| Plain radiographs | | X | | | | | X | X |
| Concomitant medication | X | X | | | | | | |
| Routine follow-up (via phone)¶ | | | | | | X | | |
| AE assessment¶ | | | X | X | X | X | X | X |

*The informed consent form must be signed before any study-related procedure.
†Vitamin D and calcium will be given daily in standard dosage from day 2 for 1 month.
‡Randomisation has to be performed as close as possible prior to the first IMP infusion.
§Administration of IMP/placebo will be given the day after surgery.
¶AE/SAE will be collected via a questionnaire at visit five and personal interview at visit four. At visits six, seven and eight, only SAE will be collected via a questionnaire. Reminders will be given by phone.
AE, adverse event; HOOS, Hip dysfunction and Osteoarthritis Outcome Score; IMP, investigational medical product; KOOS, Knee injury and Osteoarthritis Outcome Score; SAE, severe adverse event; SF-36, 36-Item Short Form Health Survey; X-ray, plain radiography.

## Statistical analysis plan

Statistical analysis of the primary endpoint will be carried out using a mixed model repeated measurements, Analysis of Variance (ANOVA), of the changes in KOOS/HOOS from baseline until 3-year follow-up with covariate adjustment for baseline values of HOOS/KOOS, and with implant type (hip and knee, cemented and uncemented) and age (continuous) as further covariates. As supportive endpoints, we will analyse HOOS/KOOS subscales at 6-year follow-up, signs of radiographic loosening at 3 years and 6 years and RAND/SF-36 at 1 year, 3 years and 6 years. We will also perform subgroup analyses of men and women and implant types.

To reduce the risk of bias during interpretation, blinded results from the analyses (with study groups labelled as group A and group B) will be presented to all the authors, who will agree in writing on two alternative interpretations.[38] Thereafter, the data manager will break the randomisation code.

As part of an exploratory analysis and to be able to define the clinical impact of our results, we will perform a responder analysis comparing the proportion of patients who achieve a substantial clinical improvement on the subscale pain in HOOS/KOOS between the treatment and the control group.[39] HOOS and KOOS values depend on age, Body Mass Index (BMI) and sex, which will be considered as cofactors in the analysis.

No interim analysis will be performed.

## Safety

### Concomitant drug treatment

After the infusion, oral supplements of vitamin D and calcium will be given once daily to both groups for the first postoperative month to prevent bisphosphonate-induced hypocalcaemia.

### Zoledronic acid

Repeated infusions of zoledronic acid has been associated with a slight increase in atrial fibrillation in the highest age groups[40] and because of this we have set an upper age limit for inclusion to 80 years. Bisphosphonate use is strongly associated with osteonecrosis of the jaw. This is, however, a very rare condition and only associated with multiple dosing over time.[41] Ten per cent of patients treated with zoledronic acid have reported acute-phase reactions,[42] which can lead to a prolonged hospital stay. Even though zoledronic acid has not been reported to cause hypocalcaemia in osteoporosis treatment,[43] normal preoperative calcium and vitamin D levels are required for inclusion for safety reasons and patients will get oral supplements for the first postoperative month. Zoledronic acid can affect kidney function and the manufacturer recommends against its use in patients with creatinine clearance <35 mL/min.[44] Bisphosphonates are used in large scale to treat patients with osteoporosis, and its safety is extensively documented.[41] Furthermore, in this study, only one dose of 4 mg[17] is given, compared with the repeated dosing of 5 mg in osteoporosis treatment.

## Adverse events

Patients will be followed-up for 6 weeks after the infusion for adverse events (AEs). The physiotherapist will record AEs at the 6-week follow-up (table 1). An AE is defined in the International Conference on Harmonisation (ICH) Guidelines for good clinical practice as 'any untoward medial occurrence in a patient or clinical investigation subject administered a pharmaceutical product and does not necessarily have a causal relationship with this treatment'. The occurrence of atypical femoral fractures and osteonecrosis of the jaw will be recorded throughout the whole study period. Serious adverse event (SAE) is defined as an AE that is fatal, life threatening, requires in-patient hospitalisation or prolonged hospitalisation, results in persistent or significant disability/incapacity or other significant medical hazards. Adverse drug reaction is defined as all untoward and unintended response to a medical product related to any dose administered and will also be recorded during the first 6 weeks. The occurrence of AE and SAE will be presented descriptively. For any harm caused through study participation, all patients are covered by the national Swedish patient insurance, Landstingens Ömsesidiga Försäkringsbolag.

## ETHICS AND DISSEMINATION
### Ethics

The study will be conducted in agreement with the Declaration of Helsinki and ICH guidelines will be adhered to. The study will be monitored by Forum Östergötland, which is part of the national organisation for clinical studies in Sweden, Forum Sydost. All completed questionnaires will be kept secured from unauthorised access within the research nurses' facility. The data for the purpose of statistical analyses will be collected in digitised files. Other data will be stored in the patients' ordinary medical chart. The study has been approved by the Regional Ethical Review Board in Linköping (DNR 2015/286-31).

### Dissemination

The results of this study will be submitted for publication in an international peer-reviewed scientific paper regardless of whether the results are positive, negative or inconclusive regarding the hypothesis of the study.

### Patient and public involvement

No patient organisation or patient representatives were involved in the design of the study. The results of our study will be disseminated to patient organisations and the public through the Swedish Orthopaedic Association and the Swedish National Joint Arthroplasty Register.

## DISCUSSION
### Strengths and limitations
#### Strengths

The main strength of this study is its size and design. To the best of our knowledge, this is the largest RCT designed to elucidate if bisphosphonates can improve outcome in

primary THA and TKA. If we can demonstrate a significant increase in patient satisfaction after bisphosphonate administration, it could revolutionise the perioperative care of patients undergoing TJR.

## Limitations

The main limitation of this study can be considered its primary patient-reported outcome measure. A hard endpoint as a prospectively collected rate of revision would be preferable, however, not feasible in this research question. Also, not all patients with loosening of their implants undergo revision surgery. Some are too old and fragile, and others have moderate symptoms and might decide to abstain from surgery. The use of a patient-reported primary endpoint will increase the relevance of the findings to patients. Improved patient-reported outcomes after TJR might in fact be more important for the majority of the patients undergoing TJR compared with prosthetic loosening assessed on radiographs, which is a secondary outcome. If we fail to demonstrate a significant increase in patient-reported outcomes, we might be able to show a decrease in radiographic signs of early loosening, which strongly correlates with late aseptic loosening.[13 14] Also, dual-energy X-ray absorptiometry will not be performed.

**Author affiliations**
[1]Department of Orthopaedic Surgery, Capio Specialistvård Motala, Motala, Sweden
[2]Department of Clinical sciences, Lund University, Lund, Sweden
[3]Department of Sports Science and Clinical Biomechanics, Syddansk Universitet Det Sundhedsvidenskabelige Fakultet, Odense, Denmark
[4]Department of Orthopaedic Surgery, Linköping University Hospital, Linköping, Sweden
[5]Department of Biomedical and Clinical Sciences, Faculty of health Sciences, Linköping University, Linköping, Sweden
[6]Wallenberg Centre for Molecular Medicine, Linköping University, Linköping, Sweden

**Acknowledgements** We are grateful for the research-friendly leadership of the healthcare providers Capio Specialistvård Motala and Aleris Specialistvård Motala. We also thank all personnel at Motala hospital for their extra effort to make this large research project possible, in specific our research nurses Anneli Kvarnestedt and Carmen Henriksson. The Knut and Alice Wallenberg foundation is acknowledged for generous support.

**Contributors** JB and HL: writing of manuscript and study design. JR and ER: revision of manuscript and study design. PA: study design and preparation of manuscript. JS: writing and revision of manuscript and study design.

**Funding** This academic drug trial is run without any support from the industry. Financial support was received from the Swedish Research Council, grant number 2014–07284, and ALF-grants from Region Östergötland, Sweden.

**Competing interests** None declared.

**Patient and public involvement** Patients and/or the public were not involved in the design, or conduct, or reporting, or dissemination plans of this research.

**Patient consent for publication** Not required.

**Provenance and peer review** Not commissioned; externally peer reviewed.

**ORCID iDs**
Jonathan Brandt http://orcid.org/0000-0002-4343-4808
Ewa Roos http://orcid.org/0000-0001-5425-2199

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
