## [Reviewer comments · BMJ Open]

ARTICLE DETAILS

TITLE (PROVISIONAL)	A single postoperative infusion of zoledronic acid to improve patient reported outcome after hip or knee replacement: study protocol for a randomized, controlled, double-blinded clinical trial.
AUTHORS	Brandt, Jonathan; Ledin, Håkan; Ranstam, Jonas; Roos, Ewa; Aspenberg, Per; Schilcher, Jörg

VERSION 1 – REVIEW

REVIEWER	Gerasimos Evangelatos Rheumatology Unit, First Department of Propaedeutic Internal Medicine, National and Kapodistrian University of Athens, Laiko General Hospital, Athens, Greece
REVIEW RETURNED	09-Jun-2020

GENERAL COMMENTS	General Comments to Authors: This is a well-written and meticulous protocol. The study design is excellent. However, I disagree with the selection of subjective primary outcomes. BMD measurement before zoledronic acid is lacking. Study timeline is not clearly presented Page 4 Line 19: Use “placebo-controlled” instead of “placebo controlled” Page 5 Line 11: “For the surgeon and THE patient” Page 5 Line 24: Instead of “creates” I would use “would possibly create” Page 5 Line 48: The time schedule of the study is demonstrated well, but the exact timeline for initiation and termination of the study needs to be clarified. Page 6: One exclusion criterion is “use of drugs that affect the bone” (corticosteroids included) and some lines later you write that “Regular use of corticosteroids more than 5 mg dexamethasone per day”. You should correct it. Furthermore, why did you choose 5mg dexamethasone and not prednisolone? Explain Page 6 Line 59: Why have you chosen 4mg zoledronic acid and not 5mg, which is the standard treatment for osteoporosis? Explain Page 7 Line 16: In the sentence “...reported not only less migration...”, which kind of migration do you refer to? Page 7: Authors will use HOOS and KOOS difference between zoledronic acid and placebo groups as primary outcome. I disagree that a subjective questionnaire, especially the subscale “pain”, would be a primary outcome for a pharmaceutical intervention, as some confounding factors can influence the results (other joint osteoarthritis, fibromyalgia, depression, etc). This is also pointed in the “Limitations” by the authors. I would prefer radiographic loosening (even if it does not lead to reoperation) as a more objective primary outcome. Page 8 Line 14: Another drawback of this protocol is that there is not defined the minimal important change for KOOS questionnaire. From a clinical point of view, this makes difficult the interpretation of
---

	borderline statistical significance. Page 9 Line 32: I suggest that “gender” is also used as a categorical covariate that could influence HOOS and KOOS pain scores. Page 10 Lines 9-10: Better use “Acute-Phase Reaction” instead of “influenza-like symptoms” as the former is more commonly used in the literature Page 10 Line 21: Prefer “Patients will be followed-up for 6 weeks after the infusion for adverse events (AE)...” than “Adverse events (AE) are followed for 6 weeks after the infusion...” Page 10 Lines 51-52: I would prefer “submitted for publication” rather than “published” Page 11: In “Limitations”, authors should also point that a dual-energy x-ray absorptiometry (DXA) did not precede the zoledronic acid infusion. It is known that the lower the baseline bone mineral density (BMD) is, the bigger is the increase of BMD. This could possible influence the bone formation around the implant
--	---

REVIEWER	Hannu T Aro Turku University Hospital and University of Turku, Finland
REVIEW RETURNED	19-Jun-2020

GENERAL COMMENTS	Background This trial deals with an important unsolved clinical issue. Population-based register studies have suggested an improved survival of hip and knee implants in patients receiving bisphosphonate treatment, but there are no randomized trials. It is still unknown if these associations are truly causal. Relevance to perform a RCT While the investigators are to be commended for this proposal, several key features limit the relevance of the protocol. A randomized clinical trial (RCT) is indicated when appropriate, practical, and ethical. A RCT is often not a realistic option for evaluation an intervention that is designed to prevent a rare outcome occurring over a long follow-up period, such as implant failure after THA. The protocol is unclear for the basic mechanism-based hypothesis. What is the expected treatment effect of zoledronic acid treatment? Is the same for cemented and uncemented implant components in different anatomic locations? Is it relevant to compare different types of total joint arthroplasties (including cemented, cementless and hybrid THAs as well as cemented and cementless TKAs) in a RCT? Review of the literature The review of literature is short and do not clearly open the view whether a bisphosphonate treatment is capable and safe in improving of implant survival. The literature review must be concise but it would be relevant to review certain issues: (1) the presence or absence of cutoff values for clinically significant early migrations of different implants, (2) the treatment effect of bisphosphonate treatment in THAs and TKAs in terms of the magnitude of reduced implant component migration (under or above the defined cutoff values), and
--

(3) the observed differences of PROMs in all RCTs which have studied the effects of antiresorptive drugs on migration of THA and TKA implants.

The trial will be based on the use of a single dose of zoledronic acid. Is a single dose enough? A short 6-month postoperative treatment period of clodronate reduced implant migration still at 4 years (Hilding & Aspenberg Acta 2006), but a population-based study has shown a dose response association between bisphosphonates use and the risk of revision (Thillemann et al. Bone 2010). Is the immediate postoperative infusion of zoledronic acid safe? Is there an increased risk of revision due to deep infections (Thillemann et al. Bone 2010)?

Power analysis

Based on the protocol, this study is designed to be adequately powered to detect a clinical difference. However, the clinical effect size used within the power calculation remains vague, resulting this study being more an exploratory study. Even as such, the trial brings unmatched new data.

Based on the current knowledge, the efficacy of antiresorptive treatment has been shown only for cemented tibial components and uncemented and cemented acetabular cups. Thus, the treatment effect of zoledronic acid seems to be different in cemented and uncemented hip and knee implants, which makes the expected treatment scenario complex and the power analysis difficult. In the current protocol, the power analysis expects a similar response in cemented and uncemented hip and knee implants.

As stated by the investigators, the main limitation of this study is its primary patient-reported outcome measure. One may argue that the selection of PROMs as the outcome measure of the treatment response of zoledronic acid is not relevant for the lack of convincing data. A recent trial of zoledronic acid (Aro J Biomech 2018) showed no differences of PROMs or objectively measured functional outcomes in a 9-year follow-up of cementless THA patients. Correctly, as discussed by the investigators, late aseptic loosening may show up in deterioration of PROMs, but are there published studies, which have shown the responsiveness of PROMs in detection of emerging implant loosening?

The use of PROM scores as the main outcome measure is largely based on the results of one trial (Friedl et al. JBJS 2009), which was not based RSA. As cited by the investigator, in the cited trial the postoperative increase of Harris hip score was significantly more pronounced in the zoledronate-treated group than in the control group. The median HHS score was 96 in the control group and 100 in the zoledronic acid group. This was informed to be statistically highly significant ($p < 0.001$) but was there a clinically meaningful difference? To the best knowledge, previous RSA-based trials (demonstrating the inhibition of stem migration by means of bisphosphonates) have shown concurrent improvements of PROMs.

Safety issues

Zoledronic acid, the most potent bisphosphonate, may cause severe

	hypocalcemia, which has been reported after hip arthroplasty (Won-Seok Do et al. J Bone Metab. 2012). Impaired renal function and vitamin D deficiency are the most significant risk factors in hypocalcemia occurrence. Vitamin D status should be assessed during the trial screening process and the list of exclusion criteria should include vitamin D deficiency. Vitamin D loading before the infusion of zoledronic acid may lower the risk of hypocalcaemia (Lyles et al. N Engl J Med. 2007). Therefore, the early start (during screening) of D-vitamin and calcium substitution is recommended. The treatment effect of the single infusion of zoledronic acid lasts for a minimum of one year. A bisphosphonate treatment always requires a concomitant D-vitamin and calcium substitution. Therefore, it is reasonable to continue D-vitamin and calcium substitution at least for one year. The timing of the infusion is important. Hip arthroplasty procedure per se has resulted in postoperative hypocalcaemia, forcing to delay the postoperative infusion of zoledronic acid (Aro et al. J Biomech 2018). Therefore, the immediate infusion of the drug (without controlling serum level of calcium) may carry a risk.
--	--

VERSION 1 – AUTHOR RESPONSE

Reviewer: 1

Reviewer Name: Gerasimos Evangelatos

Institution and Country: Rheumatology Unit, First Department of Propaedeutic Internal Medicine, National and Kapodistrian University of Athens, Laiko General Hospital, Athens, Greece

Please state any competing interests or state 'None declared': None declared

General Comments to Authors: This is a well-written and meticulous protocol. The study design is excellent. However, I disagree with the selection of subjective primary outcomes. BMD measurement before zoledronic acid is lacking. Study timeline is not clearly presented

RESPONSE: Roughly 800 patients of the planned 1000 are already included, which unfortunately precludes major changes in the protocol such as the primary outcome or the addition of BMD measurements. The proposed timeline is now presented, see p 4.

Page 4 Line 19: Use “placebo-controlled” instead of “placebo controlled”

RESPONSE: The wording is now changed according to the reviewer’s suggestion, see p 2.

Page 5 Line 11: “For the surgeon and THE patient”

RESPONSE: The wording is now changed according to the reviewer’s suggestion, see p 3.

Page 5 Line 24: Instead of “creates” I would use “would possibly create”

RESPONSE: The wording is now changed according to the reviewer’s suggestion, see p 3.

Page 5 Line 48: The time schedule of the study is demonstrated well, but the exact timeline for initiation and termination of the study needs to be clarified.

RESPONSE: We have now added the date of study initiation and state a rough estimate for the end of follow-up for the primary outcome, see p 4.

Page 6: One exclusion criterion is “use of drugs that affect the bone” (corticosteroids included) and some lines later you write that “Regular use of corticosteroids more than 5 mg dexamethasone per day”. You should correct it. Furthermore, why did you choose 5mg dexamethasone and not prednisolone? Explain

RESPONSE: Thank you for pointing out this erroneous description. This is now corrected, see p 4.

Page 6 Line 59: Why have you chosen 4mg zoledronic acid and not 5mg, which is the standard treatment for osteoporosis? Explain

RESPONSE: We used the same dose as Friedl et al.¹ Zoledronate binds with high affinity to mineral exposed to the blood stream (the traumatized peri-implant bone bed) and a lower dose compared to the standard dose in osteoporosis treatment is not counterintuitive. Also, 4mg seemed to have given a satisfactory effect in the Friedl trial and at the same time might limit side effects compared with the higher dose of 5mg. We now reference the Friedl study after the dose is given in the methods section, see p 5.

Page 7 Line 16: In the sentence "...reported not only less migration...", which kind of migration do you refer to?

RESPONSE: Thank you for pointing out this potential source of misunderstanding. We have now added the word "implant" to make clear which kind of migration we refer to. We have also added a more thorough description of implant migration in the background section.

Page 7: Authors will use HOOS and KOOS difference between zoledronic acid and placebo groups as primary outcome. I disagree that a subjective questionnaire, especially the subscale "pain", would be a primary outcome for a pharmaceutical intervention, as some confounding factors can influence the results (other joint osteoarthritis, fibromyalgia, depression, etc). This is also pointed in the "Limitations" by the authors. I would prefer radiographic loosening (even if it does not lead to reoperation) as a more objective primary outcome.

RESPONSE: We agree with the reviewer's concern that patient reported outcome might be confounded. However, we find radiographic evaluation of implant loosening unfeasible for several reasons. Radiographic loosening is typically defined as an implant that migrates to such an extent that it is visible for the human eye on plain radiographs, the component changes position, the component or cement breaks or clear lucencies occur in the bone around the implant. These radiographic signs should always be accompanied with a clinical examination and patient history to verify the diagnosis. Also, radiographic signs of loosening occur roughly 4-5 years before revision surgery² and therefore would require an extended follow-up and probably a larger sample size. This would be very close to actual revision surgery as the final outcome in terms of follow-up and therefore unfeasible for our academic RCT with limited economic resources. The use of radiolucent zones³ to predict loosening appears not reliable enough⁴ to be used as a primary outcome variable in an RCT.

*The main rationale in this trial is that the patient's subjective perception of discomfort might be more specific than early signs of radiographic loosening such as radiolucent zones, which are difficult to identify reproducibly and vary depending on the anatomic location of the implant and the method of fixation. PROMs and in specific pain are a good predictor of revision surgery⁵ and implant survival.⁶⁻⁹ Since pain is the main indication for surgery it seems reasonable to also follow pain to evaluate outcome. Also, pain is the primary outcome measure recommended by the **Outcomes Measures in Rheumatology, Workgroup Total Joint Replacement**, which is considered a strong consensus statement.*

Nonetheless, we share the reviewers concerns of a confounded primary outcome variable which is why we use radiographic loosening as a secondary outcome variable, despite our concerns mentioned above.

Page 8 Line 14: Another drawback of this protocol is that there is not defined the minimal important change for KOOS questionnaire. From a clinical point of view, this makes difficult the interpretation of borderline statistical significance.

RESPONSE: The study is powered for a statistically significant difference in the subscale pain of HOOS and KOOS between the two groups. If there is no statistically significant difference, our results will not confirm our hypothesis and therefore will not be reported as such. In that case our secondary endpoints will serve as supportive, exploratory or explanatory evidence in hierarchical order as described in the protocol. This interpretation of results is based on European Medicines agency guidelines.

Page 9 Line 32: I suggest that "gender" is also used as a categorical covariate that could influence HOOS and KOOS pain scores.

RESPONSE: We will use gender as a covariate in an explorative analysis but not the primary confirmative analysis. The addition of gender as a covariate would lead to a loss in the degrees of

freedom. If we would have expected gender to play a major role in the outcome, we would have added this as a stratification factor in the randomization. Having included more than 800 patients, such a change in protocol is not feasible.

Page 10 Lines 9-10: Better use “Acute-Phase Reaction” instead of “influenza-like symptoms” as the former is more commonly used in the literature

RESPONSE: Thank you for pointing this out. We have now changed the wording according to the reviewer’s suggestion, see p 8.

Page 10 Line 21: Prefer “Patients will be followed-up for 6 weeks after the infusion for adverse events (AE)...” than “Adverse events (AE) are followed for 6 weeks after the infusion...”

RESPONSE: This has now been changed according to the reviewer’s suggestion, see p 9.

Page 10 Lines 51-52: I would prefer “submitted for publication” rather than “published”

RESPONSE: This has now been changed according to the reviewer’s suggestion, see p 9.

Page 11: In “Limitations”, authors should also point that a dual-energy x-ray absorptiometry (DXA) did not precede the zoledronic acid infusion. It is known that the lower the baseline bone mineral density (BMD) is, the bigger is the increase of BMD. This could possibly influence the bone formation around the implant.

RESPONSE: This has now been changed according to the reviewer’s suggestion see p 10.

Reviewer 2

Background

This trial deals with an important unsolved clinical issue. Population-based register studies have suggested an improved survival of hip and knee implants in patients receiving bisphosphonate treatment, but there are no randomized trials. It is still unknown if these associations are truly causal.

Relevance to perform a RCT

While the investigators are to be commended for this proposal, several key features limit the relevance of the protocol. A randomized clinical trial (RCT) is indicated when appropriate, practical, and ethical. A RCT is often not a realistic option for evaluation an intervention that is designed to prevent a rare outcome occurring over a long follow-up period, such as implant failure after THA. The protocol is unclear for the basic mechanism-based hypothesis. What is the expected treatment effect of zoledronic acid treatment? Is the same for cemented and uncemented implant components in different anatomic locations? Is it relevant to compare different types of total joint arthroplasties (including cemented, cementless and hybrid THAs as well as cemented and cementless TKAs) in a RCT?

RESPONSE: We aimed to design a pragmatic RCT and choose patient reported outcome 3 years after a primary joint replacement as the primary outcome to achieve generalizability to allow implementation in clinical practice. The hypothesis is that aseptic implant loosening in the majority of cases is a progressive mechanism that starts with the surgery rather than as a sudden event after many years. We based this hypothesis on a strong association between increased movements of the prosthetic component in its bone bed within the first 2 years after surgery – so called “micromotion” (in the magnitude of roughly a millimetre) – and an increased risk of revision for aseptic loosening after about 10 years. This mechanism is independent of the type of fixation cemented/uncemented and the anatomic location of the prosthesis. Micromotions cannot be seen on standard radiographs and secondary signs of loosening (osteolysis and sclerosis) are typically not present in the early phases even if an implant shows increased migration.⁴ Micromotions might occur because the bone around a prosthesis is resorbed as a result of the surgical trauma, meaning that the bone-bed becomes weaker in the post-operative period due to osteoclast driven bone resorption. The hypothesis is that bisphosphonates inhibit this bone resorption through their inhibitory effect on osteoclasts and the prosthesis therefore remains more stable –less micromotion occurs. One study has found not only an association between bisphosphonate treatment and decreased micromotion but also higher functional scores on the Harris Hip Score.¹ This might indicate that micromotions leading

to later prosthetic loosening might be perceived as a minimal discomfort by patients very early after the surgery. Such a mechanism is supported through results from register-based follow-up studies using PROM's to predict implant survival. Previous RCT's evaluating the effect of bisphosphonate treatment have not detected such an improvement in patient reported outcome most likely due to small numbers in the RCT's. Sample sizes in RCT's using migration analysis are typically powered to detect differences in implant migration not PROM's. The one exception is the study by Friedl et al. Patients in that study had joint replacement due to avascular necrosis of the femoral head and not degenerative arthritis and patients were younger than the average arthritis patient that receives a joint replacement. Therefore, the study by Friedl et al has limited external validity. In our pragmatic RCT, we include the diversity of joint replacements that are chosen to meet the needs of different groups of patients.

Review of the literature

The review of literature is short and do not clearly open the view whether a bisphosphonate treatment is capable and safe in improving of implant survival.

RESPONSE: We agree, the background needs to be short and concise. Based on the reviewer's suggestion we now elaborate the background more clearly, see p 3.

The literature review must be concise but it would be relevant to review certain issues:

(1) the presence or absence of cutoff values for clinically significant early migrations of different implants

(2) the treatment effect of bisphosphonate treatment in THAs and TKAs in terms of the magnitude of reduced implant component migration (under or above the defined cutoff values), and

RESPONSE: We agree with the reviewer that this would be a very interesting topic to review.

However, the topic is complex and there are many aspects to consider which is summarized in several previous publications for hip and knee implants. Pijls et al. for example recently evaluated 53 RSA studies in total knee replacement.¹⁰ Because we do not use the technique of implant migration measurements in the current study, we believe an in-depth description of implant migration and thresholds would be a somewhat overwhelming piece of information for the reader of a study protocol. However, we now give some more information on the thought process that led to the study hypothesis based on implant migration, bisphosphonate treatment and implant survival.

(3) the observed differences of PROMs in all RCTs which have studied the effects of antiresorptive drugs on migration of THA and TKA implants.

RESPONSE: For the sake of brevity we have condensed this information to a statement saying that all but one previous trial did not show a significant improvement in PROM's related to bisphosphonate treatment, but none of these studies was powered to show such a difference.

The trial will be based on the use of a single dose of zoledronic acid. Is a single dose enough?

RESPONSE: Several previous studies have used one dose successfully. This is reasonable because the osseointegration processes occurs early after the implantation and bisphosphonate is likely to bind in high concentration to the mineral that becomes exposed during surgery. Timing is likely to be more important than the number of doses.

A short 6-month postoperative treatment period of clodronate reduced implant migration still at 4 years (Hilding & Aspenberg Acta 2006), but a population-based study has shown a dose response association between bisphosphonates use and the risk of revision (Thillemann et al. Bone 2010). Is the immediate postoperative infusion of zoledronic acid safe? Is there an increased risk of revision due to deep infections (Thillemann et al. Bone 2010)?

*RESPONSE: To our understanding the study by Thillemann et al. is a register-based study where bisphosphonate treatment was captured in Danish registries and based on redeemed prescriptions, not actual drug use. Compliance of **daily** oral bisphosphonates is reported as low as 25% and 35% after 1 year and barely better with **weekly** dosing.¹¹ Also, patients receiving bisphosphonate prescriptions in that study showed an overrepresentation of sicker patients with higher comorbidity indices ($p=0.009$) and a higher number of patients receiving THA related to fracture (50% compared to 40%). Both of these factors are known risk factors for revision due to infection after primary THA*

in Denmark.¹² Furthermore a causal relationship between bisphosphonate treatment and infection in that study is unlikely because this relationship occurred only after the **shortest** treatment duration, DDD 14-120, but not in patients with treatment durations > DDD 121 and a **reduced** risk for revision due to infection was reported in the unadjusted model for patients with > 240 DDD. An increased risk for infection in that study is likely based on undetected bias. Also, the risk estimates are based on few patients (n=72). The study by Thillemann investigated oral bisphosphonate treatment given once daily or once weekly, and zoledronic acid is administered intravenously at one time point. Because of these differences in dosing regimens the effect on the risk of infection might be different. In our study roughly 800 patients have been included and five patients with deep surgical site infections have been identified so far. These five patients yield an infection rate 0.66%, which is slightly lower than the average rate of infection at the main study site (Motala hospital). Four of these five patients with deep surgical site infections have been treated with debridement, antibiotics and implant retention, and one patient has undergone successful 2-stage implant revision. We do not know which treatment group these patients belong to, but the infection rate itself has not raised a safety concern and treatment therefore has not been unblinded.

Power analysis

Based on the protocol, this study is designed to be adequately powered to detect a clinical difference. However, the clinical effect size used within the power calculation remains vague, resulting this study being more an exploratory study. Even as such, the trial brings unmatched new data.

Based on the current knowledge, the efficacy of antiresorptive treatment has been shown only for cemented tibial components and uncemented and cemented acetabular cups. Thus, the treatment effect of zoledronic acid seems to be different in cemented and uncemented hip and knee implants, which makes the expected treatment scenario complex and the power analysis difficult. In the current protocol, the power analysis expects a similar response in cemented and uncemented hip and knee implants.

RESPONSE: We agree that there is a difference in the treatment effect in different implants, types of fixation and location of the implant. However, patients that perceive a discomfort from a joint replacement do not make a difference whether the discomfort is related to the acetabular or the femoral component and whether it is cemented or not. While uncemented tibial components are not explicitly excluded, the use of this type of fixation is very rare at both centres. So far, three patients with uncemented tibial components have been included.

Our study is designed to be pragmatic and our intention is to show on overall effect of bisphosphonate treatment on patient reported outcome, which is a well-established predictor of implant survival after joint replacement.⁶⁻⁹ To exclude certain types of implants would make the idea of generalizability of our results difficult. Based on existing register studies there seems to be a general positive effect on implant survival after bisphosphonate treatment, and studies using implant migration have shown a positive effect on migration only for some implants. In our interpretation these are not contradictory findings. Register studies show a general affect, while migration studies show a specific effect for one implant. It is not surprising that the effect of bisphosphonates in migration studies is depending not only on the treatment group but also on the methodology, the implants (e.g. polished stems subside more than matte), the type of cement, cementing technique, surgical technique and many other factors.

As stated by the investigators, the main limitation of this study is its primary patient-reported outcome measure. One may argue that the selection of PROMs as the outcome measure of the treatment response of zoledronic acid is not relevant for the lack of convincing data. A recent trial of zoledronic acid (Aro J Biomech 2018) showed no differences of PROMs or objectively measured functional outcomes in a 9-year follow-up of cementless THA patients. Correctly, as discussed by the investigators, late aseptic loosening may show up in deterioration of PROMs, but are there published studies, which have shown the responsiveness of PROMs in detection of emerging implant loosening?

RESPONSE: Existing RCT's used either BMD or migration in calculations of sample size. These are endpoints with little variation that can be measured with high precision. Sample sizes therefore are small and the lack of differences in secondary outcomes such as PROM's therefore not surprising. Our rationale to choose PROM's as the primary outcome is the finding that bisphosphonate users in large register-based studies show a better long-term implant survival rate compared to none-users. Because implant loosening is a process that often progresses over several years, patients with

loosening are likely to have clinical or subclinical symptoms before they finally undergo implant revision. It is these subclinical signs of implant loosening that we are interested in. Different PROM's have shown to predict implant loosening (see above).

The use of PROM scores as the main outcome measure is largely based on the results of one trial (Friedl et al. JBJS 2009), which was not based RSA. As cited by the investigator, in the cited trial the postoperative increase of Harris hip score was significantly more pronounced in the zoledronate-treated group than in the control group. The median HHS score was 96 in the control group and 100 in the zoledronic acid group. This was informed to be statistically highly significant ($p < 0.001$) but was there a clinically meaningful difference? To the best knowledge, previous RSA-based trials (demonstrating the inhibition of stem migration by means of bisphosphonates) have shown concurrent improvements of PROMs.

RESPONSE: All previous trials were powered to detect differences in migration, not PROMs. The absence of a statistically significant difference in PROMs in these trials therefore is not unexpected. The minimal important change is often reported to be 8-10 points but this estimate is dependent on contextual factors such as patient age, intervention and time to follow-up, which complicates the formulation of a generic value of the minimal important change for any patient reported outcome measure. Our trial is powered to detect a difference of 3 points between the groups, based on a consensus decision within the research team, taking all available evidence into account at the time of study design in 2014. With a sample size of 500 patients per group, this trial is much larger than any previous trial performed, and our hypothesis is to detect a subclinical improvement in HOOS and KOOS which is likely coupled to a lower risk of revision in the future. We have now added a clarification of the decision process on page 8 in the protocol.

Safety issues

Zoledronic acid, the most potent bisphosphonate, may cause severe hypocalcemia, which has been reported after hip arthroplasty (Won-Seok Do et al. J Bone Metab. 2012). Impaired renal function and vitamin D deficiency are the most significant risk factors in hypocalcemia occurrence. Vitamin D status should be assessed during the trial screening process and the list of exclusion criteria should include vitamin D deficiency.

Vitamin D loading before the infusion of zoledronic acid may lower the risk of hypocalcaemia (Lyles et al. N Engl J Med. 2007). Therefore, the early start (during screening) of D-vitamin and calcium substitution is recommended. The treatment effect of the single infusion of zoledronic acid lasts for a minimum of one year. A bisphosphonate treatment always requires a concomitant D-vitamin and calcium substitution. Therefore, it is reasonable to continue D-vitamin and calcium substitution at least for one year.

RESPONSE: We thank the reviewer for these important comments. While we agree in general, we do not perceive the necessity for Vitamin D and calcium substitution quite as clear. Vitamin D and Calcium are typically given to prevent fractures not to treat hypocalcemia, and whether to substitute or not remains a matter of discussion.¹³ One very recent zoledronate trial only used Vit D not calcium concomitant to zoledronate treatment.¹⁴ Also in the first pivotal trial if zoledronate, all noted cases of hypocalcaemia were described as transient and asymptomatic.¹⁵ At the time of trial initiation, we followed recommendations of the Summary of Product Characteristics for zoledronic acid provided by the European Medicines Product Agency. Additional to that we followed all advice given by the Swedish Medical Products Agency and the Swedish National Ethical Review Authority during the application for the RCT. So far roughly 800 patients are included, and no adverse events related to hypocalcemia have been noted. Also, no exclusions due to preoperative hypocalcemia have been done, and all patients with impaired renal function and malignancies are excluded from the trial.

The timing of the infusion is important. Hip arthroplasty procedure per se has resulted in postoperative hypocalcaemia, forcing to delay the postoperative infusion of zoledronic acid (Aro et al. J Biomech 2018). Therefore, the immediate infusion of the drug (without controlling serum level of calcium) may carry a risk.

RESPONSE: The occurrence of hypocalcemia after total joint replacement appears to vary during the postoperative period and shows a strong correlation with intraoperative bleeding and fluid substitution.¹⁶ The duration is unknown and the clinical impact unclear. Therefore, routine evaluation is not recommended.¹⁷ Among the roughly 800 patients that have been included in our study, using 4mg zoledronic acid not the conventional 5mg, so far no adverse events related to hypocalcemia have been recorded. Also, in a recent osteoporosis trial using zoledronate, no calcium substitution

was given beyond the recommended dietary intake of 1g per day.¹⁴ Furthermore, our patients are healthier than the average patient partly because they have been selected suitable for total joint replacement and partly due to exclusion criteria related to our study.

REFERENCES

1. Friedl G, Radl R, Stihsen C, et al. The effect of a single infusion of zoledronic acid on early implant migration in total hip arthroplasty. A randomized, double-blind, controlled trial. *The Journal of bone and joint surgery American volume* 2009;91(2):274-81. doi: 10.2106/JBJS.G.01193
2. Aghayev E, Teuscher R, Neukamp M, et al. The course of radiographic loosening, pain and functional outcome around the first revision of a total hip arthroplasty. *BMC Musculoskeletal Disord* 2013;14:167. doi: 10.1186/1471-2474-14-167 [published Online First: 2013/05/16]
3. DeLee JG, Charnley J. Radiological demarcation of cemented sockets in total hip replacement. *Clinical orthopaedics and related research* 1976(121):20-32. [published Online First: 1976/11/01]
4. Abrahams JM, Kim YS, Callary SA, et al. The diagnostic performance of radiographic criteria to detect aseptic acetabular component loosening after revision total hip arthroplasty. *Bone Joint J* 2017;99-B(4):458-64. doi: 10.1302/0301-620X.99B4.BJJ-2016-0804.R1 [published Online First: 2017/04/08]
5. Bryant MJ, Kernohan WG, Nixon JR, et al. A statistical analysis of hip scores. *The Journal of bone and joint surgery British volume* 1993;75(5):705-9. [published Online First: 1993/09/01]
6. Rothwell AG, Hooper GJ, Hobbs A, et al. An analysis of the Oxford hip and knee scores and their relationship to early joint revision in the New Zealand Joint Registry. *The Journal of bone and joint surgery British volume* 2010;92(3):413-8. doi: 10.1302/0301-620x.92b3.22913 [published Online First: 2010/03/02]
7. Rolfson O, Bohm E, Franklin P, et al. Patient-reported outcome measures in arthroplasty registries Report of the Patient-Reported Outcome Measures Working Group of the International Society of Arthroplasty Registries Part II. Recommendations for selection, administration, and analysis. *Acta Orthop* 2016;87 Suppl 1:9-23. doi: 10.1080/17453674.2016.1181816 [published Online First: 2016/05/27]
8. Devane P, Horne G, Gehling DJ. Oxford hip scores at 6 months and 5 years are associated with total hip revision within the subsequent 2 years. *Clinical orthopaedics and related research* 2013;471(12):3870-4. doi: 10.1007/s11999-013-2880-3 [published Online First: 2013/03/05]
9. Singh JA, Schleck C, Harmsen S, et al. Clinically important improvement thresholds for Harris Hip Score and its ability to predict revision risk after primary total hip arthroplasty. *BMC Musculoskeletal Disord* 2016;17:256. doi: 10.1186/s12891-016-1106-8 [published Online First: 2016/06/12]
10. Pijls BG, Plevier JWM, Nelissen R. RSA migration of total knee replacements. *Acta Orthop* 2018;89(3):320-28. doi: 10.1080/17453674.2018.1443635 [published Online First: 2018/03/07]
11. Silverman SL, Schousboe JT, Gold DT. Oral bisphosphonate compliance and persistence: a matter of choice? *Osteoporosis international : a journal established as result of cooperation between the European Foundation for Osteoporosis and the National Osteoporosis Foundation of the USA* 2011;22(1):21-6. doi: 10.1007/s00198-010-1274-6 [published Online First: 2010/05/12]
12. Pedersen AB, Svendsen JE, Johnsen SP, et al. Risk factors for revision due to infection after primary total hip arthroplasty. *Acta Orthopaedica* 2010;81(5):542-47. doi: 10.3109/17453674.2010.519908
13. Yao P, Bennett D, Mafham M, et al. Vitamin D and Calcium for the Prevention of Fracture: A Systematic Review and Meta-analysis. *JAMA Netw Open* 2019;2(12):e1917789. doi: 10.1001/jamanetworkopen.2019.17789 [published Online First: 2019/12/21]
14. Reid IR, Horne AM, Mihov B, et al. Fracture Prevention with Zoledronate in Older Women with Osteopenia. *N Engl J Med* 2018;379(25):2407-16. doi: 10.1056/NEJMoa1808082 [published Online First: 2018/12/24]
15. Black DM, Delmas PD, Eastell R, et al. Once-yearly zoledronic acid for treatment of postmenopausal osteoporosis. *New England Journal of Medicine* 2007;356(18):1809-22.
16. Gai P, Sun H, Sui L, et al. Hypocalcaemia After Total Knee Arthroplasty and its Clinical Significance. *Anticancer Res* 2016;36(3):1309-11. [published Online First: 2016/03/16]

17. Wu XD, Zhu ZL, Xiao PC, et al. Are Routine Postoperative Laboratory Tests Necessary After Primary Total Hip Arthroplasty? *The Journal of arthroplasty* 2020 doi: 10.1016/j.arth.2020.04.097 [published Online First: 2020/05/30]

VERSION 2 – REVIEW

REVIEWER	Hannu Aro University of Turku, Turku, Finland
REVIEW RETURNED	08-Aug-2020

GENERAL COMMENTS	Further comments:  1. This trial has no trial registration. There is registration only with EudraCT (European Union Drug Regulating Authorities Clinical Trials). However, this is only an obligatory registration for drug trials in order to get an official registration number for local approvals. It does not represent a real trial registration (like clinicaltrials.gov) which informs all the essential data about the trial, including the trial population, interventions and endpoints. 2. Introduction does not inform the current controversial situation in the literature – the latest data failed to show that zoledronic acid is able to reduce migration of an uncemented femoral stem. 3. The use of PROM scores as the main outcome measure is based on the results of one trial (Friedl et al. JBJS 2009), which was not based radiostereometric analysis (RSA). 4. The power analysis has a potential flaw. Different types of total joint arthroplasties (including cemented, cementless and hybrid THAs as well as cemented and cementless TKAs) are combined in a RCT. However, there is no RSA data that bisphosphonates inhibits early migration of cementless femoral stems and cementless TKA.
--

VERSION 2 – AUTHOR RESPONSE

Reviewer: 2

Reviewer Name: Hannu Aro

Institution and Country: University of Turku, Turku, Finland

Please state any competing interests or state 'None declared': None declared

Further comments:

1. This trial has no trial registration. There is registration only with EudraCT (European Union Drug Regulating Authorities Clinical Trials). However, this is only an obligatory registration for drug trials in order to get an official registration number for local approvals. It does not represent a real trial registration (like clinicaltrials.gov) which informs all the essential data about the trial, including the trial population, interventions and endpoints.

RESPONSE: Please see the editorial comment.

2. Introduction does not inform the current controversial situation in the literature – the latest data failed to show that zoledronic acid is able to reduce migration of an uncemented femoral stem.

*RESPONSE: We are very thankful for the reviewer's perseverance in this matter and we agree that this controversy is interesting when migration of specific uncemented implants is discussed or studied. However, our trial does not evaluate migration patterns of specific implants, but patient reported outcome related to primary **total** joint arthroplasty of the hip and knee. Patients will report on the functional outcome of the **total** joint not only one component of the total joint arthroplasty. Having included roughly 860 of the proposed 1000 patients in the trial, roughly 15% of patients have received uncemented femoral stems, which is a relatively small part of the total study cohort. Because we*

stratify randomization based on the type of fixation (cemented or not), we will be able to perform subgroup analysis for uncemented components without the need to compensate for multiple comparisons in the statistical analysis. For the sake of brevity, we have now added one short comment to describe the unclear effect of zoledronic acid in uncemented stems together with a recent reference, page 3.

3. The use of PROM scores as the main outcome measure is based on the results of one trial (Friedl et al. JBJS 2009), which was not based radiostereometric analysis (RSA).

*RESPONSE: The use of PROM scores as the primary outcome is based on several consensus statements, of which the most important is probably the one by the **Outcomes Measures in Rheumatology, Workgroup Total Joint Replacement (1)**. We motivate the use, recognize and discuss the potential limitations of a patient reported outcome measure (PROM) as the primary outcome on page 5, in the limitation section on page 10. Also, we plan for secondary outcome measures for the scenario that we find no relevant differences for the primary outcome. We agree that the trial by Friedl is the only bisphosphonate drug trial that showed a difference in PROM, and we agree that the trial used EBRA instead of RSA. However, even if we do not use RSA data as the primary outcome nor any other measurement of migration, we find it relevant to mention the study by Friedl.*

4. The power analysis has a potential flaw. Different types of total joint arthroplasties (including cemented, cementless and hybrid THAs as well as cemented and cementless TKAs) are combined in a RCT. However, there is no RSA data that bisphosphonates inhibits early migration of cementless femoral stems and cementless TKA.

RESPONSE: Because outcomes in our trial are not based on RSA migration data but patient reported outcome, we used patient reported outcome data in our sample size calculations irrespective of RSA data. We do not agree that this makes our sample size calculations flawed.

References:

Singh JA, Dowsey MM, Dohm M, et al. Achieving Consensus on Total Joint Replacement Trial Outcome Reporting Using the OMERACT Filter: Endorsement of the Final Core Domain Set for Total Hip and Total Knee Replacement Trials for Endstage Arthritis. *The Journal of rheumatology* 2017;44:1723-6

VERSION 3 – REVIEW

REVIEWER	Hannu T Aro Turku University Hospital, Turku, Finland
REVIEW RETURNED	24-Aug-2020
GENERAL COMMENTS	No further comments.